# Oxidative Potential Characterization of Different PM_2.5_ Sources and Components in Beijing and the Surrounding Region

**DOI:** 10.3390/ijerph20065109

**Published:** 2023-03-14

**Authors:** Wei Wen, Tongxin Hua, Lei Liu, Xiaoyu Liu, Xin Ma, Song Shen, Zifan Deng

**Affiliations:** 1School of Energy and Environmental Engineering, University of Science and Technology Beijing, Beijing 100083, China; 2State Key Laboratory of Severe Weather and Key Laboratory of Atmospheric Chemistry of CMA, Chinese Academy of Meteorological Sciences, Beijing 100081, China; 3Beijing Municipal Research Institute of Eco-Environmental Protection, Beijing 100037, China; 4CMA Earth System Modeling and Prediction Centre, Beijing 100081, China

**Keywords:** PM_2.5_, PCA, OC/EC, oxidative potential, health risks, source analysis

## Abstract

With the implementation of air pollution control measures, the concentration of air pollutants in the North China Plain has exhibited a downward trend, but severe fine particulate matter (PM_2.5_) pollution remains. PM_2.5_ is harmful to human health, and the exploration of its source characteristics and potential hazards has become the key to mitigating PM_2.5_ pollution. In this study, PM_2.5_ samples were collected in Beijing and Gucheng during the summer of 2019. PM_2.5_ components, its oxidative potential (OP), and health risks were characterized. The average PM_2.5_ concentrations in Beijing and Gucheng during the sampling period were 34.0 ± 6.1 μg/m^3^ and 37.1 ± 6.9 μg/m^3^, respectively. The principal component analysis (PCA) results indicated that the main sources of PM_2.5_ in Beijing were vehicle exhaust and secondary components and that the main sources in Gucheng were industrial emissions, dust and biomass combustion. The OP values were 91.6 ± 42.1 and 82.2 ± 47.1 pmol/(min·m^3^), respectively, at these two sites. The correlation between the chemical components and the OP values varied with the PM_2.5_ sources at these two locations. The health risk assessment results demonstrated that Cr and As were potentially carcinogenic to all populations at both sites, and Cd posed a potential carcinogenic risk for adults in Gucheng. Regional cooperation regarding air pollution control must be strengthened to further reduce PM_2.5_ pollution and its adverse health effects.

## 1. Introduction

Rapid urbanization has led to an increase in the urban population density, vehicle ownership, and urban construction, which in turn has exacerbated air pollution [1]. Fine particulate matter (PM_2.5_) pollution is one of the major air pollution problems in China [2]. PM_2.5_ is defined as particulate matter with an aerodynamic diameter smaller than 2.5 μm, and its components include sulfate, nitrate, ammonium, carbonaceous components, and metallic components [3,4]. Compared to the characteristics of coarse particulate matter, PM_2.5_ is smaller in size and easier to transport, and its harmful components are more likely to enter the human body through the respiratory tract, causing health hazards [5]. Atmospheric PM_2.5_ exposure has been identified as an important risk factor of the disease burden in countries worldwide [6]. This is due to the oxidative stress triggered by PM_2.5_-induced reactive oxygen species (ROS) [7], which threaten the body’s antioxidant system and cause inflammation [8]. This represents an important potential mechanism through which PM_2.5_ affects human health [7,9,10]. The ability of PM_2.5_ to induce ROS is referred to as the oxidative potential (OP) [11,12]. This quantity is an important indicator of the health risks of atmospheric PM_2.5_. Through epidemiological and toxicological studies, it has been found that the OP of PM_2.5_ is strongly linked to human metabolic system disorders [13], fetal growth and development [14], and cardiovascular and respiratory diseases [15], such as heart disease, asthma, and lung cancer. Therefore, PM_2.5_ pollution has become a serious threat to public health and an obstacle to the healthy development of the economy and society.

As one of the most important cities in the North China Plain, Beijing is influenced by multiple pollution sources [16], and the impact of regional transport cannot be ignored [17]. In recent years, with the implementation of the Air Pollution Prevention and Control Action Plan and the Three-Year Action Plan for Winning the Blue Sky Defense War, the concentration of major pollutants in Beijing has exhibited a downward trend, and the annual average PM_2.5_ concentration has declined by 13.67% [18]. According to statistics obtained from the Beijing Municipal Ecology and Environment Bureau, the average PM_2.5_ concentration in 2019 (42 μg/m^3^) dropped by 53.1% from that in 2013 (85 μg/m^3^) [18], and the air quality was greatly improved. In September 2020, China proposed carbon peaking and carbon neutrality goals [19]. This was followed by a series of policy measures to reduce pollution and carbon emissions and mitigate the health impacts of air pollution [20]. However, previous studies can hardly support this goal. Previous studies regarding the OP of PM_2.5_ in Beijing have focused on localized areas, while the sources and composition of PM_2.5_ have considerably changed with development. Therefore, there is a need for a comparative analysis of simultaneous observation studies in Beijing and surrounding regions. Hence, understanding the contributions of different sources to the OP value is essential for implementing relevant emission reduction measures. The OP values for different components and sources must be further explored to provide support for pollution prevention and control.

Within the context of the above carbon peaking and carbon neutrality goals, this study aimed to investigate the sources of PM_2.5_ in Beijing and the surrounding region and to study its effects on the OP value. Summer PM_2.5_ monitoring and sample collection were conducted at monitoring sites in the urban areas of Beijing and suburban Gucheng, Hebei. Pearson correlation analysis (PCA) was employed to analyze the sources and concentration characteristics of the PM_2.5_ fractions at these two locations. Based on source analysis, the OP of PM_2.5_ was analyzed via the ascorbic acid (AA) method to assess the health effects of atmospheric PM_2.5_ in Beijing and Gucheng and to investigate the contribution of each component to the OP value. Additionally, a health risk assessment of several major elements in PM_2.5_ was conducted to evaluate their carcinogenic and noncarcinogenic risks to different populations. This study provides a reference basis for the development of pollution prevention and control measures and policies.

## 2. Methods

### 2.1. Site Description

The sampling sites in this study were the Chinese Academy of Meteorological Sciences (CAMS) station and the Gucheng Ecological and Agricultural Meteorological Experiment Station of the CAMS. As shown in Figure 1, one site is located on Zhongguancun South Street, Haidian District, Beijing (39.93° N, 116.32° E, 53 m), and the other site is located in Gucheng town, Dingxing County, Hebei Province (39.08° N, 115.40° E, 14.2 m). The Beijing station is located at the center of the city between the Second and Third Ring Roads. There are many buildings around the station, and the flow of vehicles is high. It is affected by residents’ emissions and motor vehicle exhaust. The Gucheng station is located in an important PM_2.5_ regional transmission channel in Beijing and can represent the contributions of surrounding areas to pollution at this site. There are farmlands, villages, and highways around the Gucheng site, and the terrain is flat and open. In this study, PM_2.5_ samples were collected from 1 July to 31 July 2019. The sampling duration was 23 h, from 9:00 a.m. to 8:00 a.m. on the next day. The Beijing station adopted a TH1000 high-flow particle sampler, and the Hebei station adopted a TH-150AII medium-flow particle sampler (Tianhong Instrument Company, Wuhan, China), with sampling flow rates of 1.05 m^3^/min and 100 L/min, respectively. The sampling filter used in the experiment was a quartz filter produced by the PALL Company. Before sample collection, the quartz filters were baked at 600 °C for 5 h. The sampled filters were wrapped with aluminum foil and stored at −18 °C for subsequent analysis. The quality control procedures included the collection of field blanks obtained by installing filters in the sampler without air flow. Field blanks were collected before and after the sampling campaign.

Meteorological data during the sampling period were obtained from the Meteorological Information Combine Analysis Process System (MICAPS). The meteorological conditions in Beijing and Gucheng during the sampling period are shown in Figure 2. The meteorological conditions during the sampling period in both places can reflect the typical characteristics of the study area in summer.

Meteoinfo was used to analyze the impact of regional transport on Beijing and its surrounding areas during the sampling period. The meteorological field data used for the study were obtained from the National Centers for Environmental Prediction (NCEP) Global Data Assimilation System (GDAS) (https://www.ready.noaa.gov/archives.php (accessed on 15 October 2022)). A 72 h backward trajectory at an altitude of 500 m was simulated at an interval of six hours.

### 2.2. Sample Analysis

#### 2.2.1. Component Analysis

Organic carbon (OC) and elemental carbon (EC) were analyzed by a multiband thermal/optical carbon analyzer (DRI Model 2015, Magee, CA, USA) with a modified-A temperature protocol [21] using a 0.495 cm^2^ filter sample.

A 2.01 cm^2^ filter sample was cut, sonicated in 10 mL of ultrapure water for 1 h at room temperature, and then filtered through a 0.45 μm polyethersulfone syringe filter. The Cl^−^, NO_3_^−^, SO_4_^2−^, NH_4_^+^, K^+^, Mg^2+^, Na^+^, and Ca^2+^ concentrations in the extracts were measured via ion chromatography (ICS3000, Thermo Fisher, Waltham, MA, USA). Anions were analyzed through an IonPac AS14 column and an AG14 guard column using a 4.5 mmol/L Na_2_CO_3_ solution and a 1.4 mmol/L NaHCO_3_ solution, respectively, at a flow rate of 1.2 mL min^−1^. Cations were analyzed through an IonPac CS12 column and a CG12 guard column using a 20 mmol/L methanesulfonic acid solution at a flow rate of 1.0 mL min^−1^. The method detection limits for Cl^−^, NO_3_^−^, SO_4_^2−^, NH_4_^+^, K^+^, Mg^2+^, Na^+^, and Ca^2+^ were 0.016, 0.024, 0.067, 0.022, 0.016, 0.015, 0.021, and 0.039 µg m^−3^, respectively.

A 2.01 cm^2^ filter sample was cut and sonicated in 4 mL of ultrapure water for 30 min. The sample solution was filtered through a 0.45 μm polyethersulfone syringe filter, and 50 μL of concentrated nitric acid was then added. Inductively coupled plasma-mass spectrometry (7900, Agilent Technologies, Santa Clara, CA, USA) was used to determine the Cr, Mn, Fe, Co, Ni, Cu, Cd, and Pb concentrations. The method detection limits for Cr, Mn, Fe, Co, Ni, Cu, Cd, and Pb were 0.021, 0.009, 0.275, 0.001, 0.004, 0.052, 0.005, and 0.003 ng m^−3^, respectively. One standard sample was analyzed for every ten samples to ensure recoveries ranging from 80 to 120%.

The chemical component data were corrected by the field blank data.

#### 2.2.2. Oxidative Potential Analysis

In this study, the AA method was used to determine the OP of the PM_2.5_ samples. A 2.01 cm^2^ piece was cut from the filter, sonicated in 4 mL of ultrapure water for 30 min and filtered through a 0.45 μm polyethersulfone syringe filter. Then, 30 μL of 10 mmol/L AA solution was added to 3 mL of the filtrate and thoroughly shaken. The remaining AA was determined via high-performance liquid chromatography (EX1700s, Wufeng Scientific Instruments, Shanghai, China) at 265 nm 5 times at 30 min intervals. The rate of AA consumption followed a linear relationship of AA versus time with a correlation coefficient greater than 0.985. The OP is expressed with volume normalization.
(1)$$\mathrm{OP}\left({\mathrm{pmol}\;\min}^{-1}m^{-1}\right)=\frac{D_{S}\left({\mathrm{pmol}\;\min}^{-1}\right)-D_{b}\left({\mathrm{pmol}\;\min}^{-1}\right)}{V_{t}\left(m^{-3}\right)\times \frac{A_{P}\left({\mathrm{cm}}^{2}\right)}{A_{t}\left({\mathrm{cm}}^{2}\right)}\times \frac{V_{r}\left(\mathrm{mL}\right)}{V_{e}\left(\mathrm{mL}\right)}}$$

D_s_ is the AA consumption rate of the samples, D_b_ is the AA consumption rate of the blank filter membrane, and V_t_ is the total volume sampled under standard conditions. A_p_ and A_t_ are the intercepted membrane area and the total membrane area, respectively. V_r_ and V_e_ are the volumes of the participating and total extraction solutions, respectively [22].

### 2.3. Source Analysis

PCA is a statistical method for correlation and analysis of variance. Matrix eigenvalues of the variance and covariance of the variables are calculated based on the initial concentrations of the components. The complex variables are downscaled to summarize the independent factors that play a major role. Based on the calculated loadings of each independent factor combined with the nature of the pollution sources of the variables, the dominant principal components are identified inductively, and the pollutant sources are analyzed qualitatively.

### 2.4. Health Risk Assessment Methods

Heavy metals contained in the composition of atmospheric PM can enter the human body through breathing, contact exposure, and other means, posing a threat to human health. Health risk assessment can be performed using the model recommended by the US Environmental Protection Agency.
(2)$$\;\mathrm{ADD}=\frac{\left(C\times \mathrm{IR}\times \mathrm{EF}\times \mathrm{ED}\right)}{\mathrm{BW}\times \mathrm{AT}}$$
(3)$$\;\mathrm{LADD}=\frac{\left(C\times \mathrm{IR}\times \mathrm{EF}\times \mathrm{ED}\right)}{\mathrm{BW}\times \mathrm{AT}}$$

ADD represents the average daily dose (mg/(kg·d)), LADD represents the life average daily dose (mg/(kg·d)), C represents the concentration of heavy metals (mg/m^3^), IR represents the inhalation rate (m^3^/d), EF represents the exposure frequency (d/a), ED represents the exposure duration (d), BW represents the body weight (kg), and AT represents the average contact time (d).

The carcinogenic risk index and noncarcinogenic risk index are calculated as follows:(4)$$\;\mathrm{HQ}=\frac{\mathrm{ADD}}{\mathrm{RfD}}$$
(5) R=LADD×SF 

HQ represents the hazard quotient; when HQ < 1, the noncarcinogenic risk is low, and when HQ ≥ 1, the exposure dose exceeds the threshold, and a noncarcinogenic risk may occur. RfD represents the reference dose (mg/(kg·d)). R also represents the reference dose, and a value of R < 1 × 10^−6^ indicates the acceptable level of the population. When 1 × 10^−6^ < R < 1 × 10^−4^, there is a potential carcinogenic risk. A value of R > 1 × 10^−4^ indicates that pollution may pose a cancer risk to the population. SF represents the carcinogenic intensity coefficient of a carcinogenic pollutant (kg·day/mg). The values of these parameters are shown in Table 1 and Table 2 (US EPA and Ministry of Ecology and Environment of the People’s Republic of China) [23,24,25,26].

## 3. Results and Discussion

### 3.1. Characteristics of PM_2.5_

#### 3.1.1. PM_2.5_ Concentration Characteristics

The ranges of PM_2.5_ concentration changes at the sites in Beijing and Gucheng during the sampling period were 7.1~79.0 μg/m^3^ and 11.6~68.5 μg/m^3^, respectively. The monthly average mass concentrations of PM_2.5_ in Beijing and Gucheng were 34.0 ± 6.1 μg/m^3^ and 37.1 ± 6.9 μg/m^3^, respectively. Figure 3 presents the variation in daily average PM_2.5_ concentrations in July 2019 at the sampling sites in Beijing and Gucheng. Beijing and Gucheng are far from each other, but the PM_2.5_ daily average concentration trends were similar. This is mainly because the PM_2.5_ pollution has regional characteristics. The average mass concentration of PM_2.5_ in Gucheng is 1.09 times higher than that in Beijing. The daily average concentration of PM_2.5_ in Gucheng during the sampling period met the ambient air quality standard level II of PM_2.5_ concentration (75 μg/m^3^), but the Beijing PM_2.5_ concentration exceeded the standard limit on 26 July (79 μg/m^3^). According to meteorological data (Figure 2) during the sampling period, the day featured high temperatures, relatively low wind speeds, and a gradual increase in humidity.

The influence of meteorological conditions on air quality cannot be ignored because they influence the concentration and distribution of various atmospheric pollutants. Therefore, this study focuses on the temperature, wind speed, and relative humidity, which affect pollution generation [27]. Figure 4 shows the fitted plots of the correlation between meteorological elements (temperature, wind speed, relative humidity) and PM_2.5_ concentrations at the two locations during the sampling period. The figure shows that temperature and relative humidity exhibit a positive correlation with the PM_2.5_ concentration at the two sites, while wind speed exhibits a more obvious negative correlation. The higher the temperature and relative humidity are, the higher the PM_2.5_ concentration. In contrast, the higher the wind speed is, the lower the PM_2.5_ pollution. Higher temperatures in summer can promote the formation of secondary aerosols, which can lead to an increase in PM_2.5_ concentrations [28,29]. Higher relative humidity is conducive to the conversion of precursor gases (SO_2_ and NO_x_) to PM_2.5_. Wind speed and wind direction are important meteorological factors affecting the accumulation, diffusion, and regional transmission of atmospheric pollutants [30]. The two sampling sites are located in the North China Plain, and the effects of these meteorological parameters on PM_2.5_ concentrations are similar to those in a previous study conducted by Yang et al. [29].

PM_2.5_ concentrations are influenced not only by local sources but also by regional transport. Data obtained from the Beijing Municipal Bureau of Ecology and Environment, covering the third round of PM_2.5_ source analysis in Beijing, indicated that regional transmission accounts for approximately 40% of the total concentration [31]. The trajectory clustering results are shown in Figure 5. Six clustering trajectories were obtained in Beijing (a), and five clustering trajectories were obtained in Gucheng (b).

The trajectories of clustered trajectories in Beijing during the sampling period were divided into six clusters. Cluster 1 involves short-range transport from the east over the Bohai Sea, Hebei, and Tianjin, which occurs frequently and accounts for 37.5% of the total trajectories. As indicated in Table 3, the PM_2.5_ concentration is 35.1 μg/m^3^, which is probably due to the transport of industrial source emissions from Hebei and Tianjin. Clusters 2 and 6 represent long-range transport from the west and northwest via Mongolia and Inner Mongolia, respectively, with relatively high PM_2.5_ concentrations of 31.0 and 27.0 μg/m^3^, respectively. This material may represent sand and dust from the arid Gobi regions of Mongolia and Inner Mongolia. Cluster 3 comes from the southeast via the Bohai Sea, Shandong, and Hebei with a PM_2.5_ concentration of 43.3 μg/m^3^. This cluster may be related to marine transport emissions. Cluster 4 involves short-range transport from southern Hebei with the highest PM_2.5_ concentration of 48.6 μg/m^3^. This trajectory passes over dense industrial cities in southern Hebei and is associated with industrial source emissions. Cluster 5 is the smallest and cleanest proportion of long-distance transport from northeastern Inner Mongolia. There are five clustered trajectories in Gucheng. Cluster 1 comes from the southeast via the Bohai Sea and Shandong, accounting for 34.82% of the total trajectories. The high PM_2.5_ concentration may originate from marine transportation pollution emissions. Cluster 2 involves long-distance transport from Inner Mongolia. Cluster 3 involves short-distance transport from southern Hebei and Shandong, accounting for 32.14% of the total trajectories. The dense industry in the Handan area in southern Hebei may lead to its high PM_2.5_ concentration. Cluster 4 is relatively clean and is from northeastern Inner Mongolia. Cluster 5 involves short-range transport from the southwest through Shanxi. The highest PM_2.5_ concentrations may originate from emissions from industrial activities such as coal mining and metallurgy in Shanxi.

#### 3.1.2. Characteristics of the Chemical Components in PM_2.5_

The average concentrations of atmospheric PM_2.5_ carbonaceous components and water-soluble ions obtained for Beijing and Gucheng are shown in Figure 6. Figure 6a shows the daily carbon component (OC and EC) and water-soluble element component in Beijing in summer, (b) shows Gucheng, and (c) shows the average concentration of the two sites. Figure 7 shows that OC, EC, and secondary ions (SO_4_^2−^, NO_3_^−^, and NH_4_^+^) exhibited high proportions in PM_2.5_ at both locations. The highest NO_3_^−^ proportion was 23.23% in Beijing, and the rest comprised SO_4_^2−^, OC, NH_4_^+^, other components and EC in this order. The highest SO_4_^2−^ content was 25.09% in Gucheng, and the rest comprised NO_3_^−^, NH_4_^+^, OC, other components and EC in this order.

OC and EC are important components of PM_2.5_. Most of the EC comes from the inadequate combustion of carbon-containing fuels. OC comes from direct emissions of primary OC and secondary OC generated by photochemical reactions [32]. The average concentrations of OC and EC in Beijing PM_2.5_ are 6.34 and 1.96 μg/m^3^, respectively. As shown in Figure 7, OC and EC account for 18.67% and 5.76% of the total PM_2.5_ concentration in Beijing, respectively. However, OC and EC accounted for 14.74% and 7.58% in Gucheng, respectively. A previous study showed that the main contributor to the OC and EC concentrations in Beijing was gasoline vehicles [33,34]. The differences in the OC and EC percentages between the two sites may originate from their different sources of PM_2.5_. The higher EC percentage in Gucheng was from the inadequate combustion of industrial coal emissions.

The SO_4_^2−^, NO_3_^−^, and NH_4_^+^ concentrations were also high at both sites. These components are mainly influenced by gaseous precursors: SO_2_, NO_x_, and NH_3_. The average concentrations of most ions in Gucheng are higher than those in Beijing. According to the report on the State of the Ecology and Environment in China in 2019, Hebei’s SO_2_ emissions are much larger than Beijing’s [18]. The high SO_2_ emissions in Hebei lead to elevated levels of SO_4_^2−^. The Hebei region has high levels of industrial coal-fired emissions, while Beijing is dominated by motor vehicle and domestic source emissions. From Figure 7, NH_4_^+^ accounted for 16.34% and 17.52% in Beijing and Gucheng, respectively. NH_4_^+^ is mostly converted from NH_3_ in the atmosphere, and NH_3_ mainly comes from the use of ammonia fertilizer in agricultural and livestock production and other emission sources, such as soil microbial ammonification processes [35]. The fertilization of farmland around Gucheng made the percentage of NH_4_^+^ slightly higher in Gucheng than in Beijing. The percentage of NO_3_^−^ is similar between Beijing and Gucheng. In recent years, the per capita ownership of motor vehicles in both Beijing and Hebei has increased rapidly, and the emission of vehicle exhaust has released a large amount of NO_x_ into the atmosphere. In addition, the implementation of the coal to gas transition policy in the Beijing–Tianjin–Hebei region has led to an increase in the concentration of NO in the atmosphere [36].

Figure 8 shows the concentrations of the water-soluble elements at the two sites. As shown in Figure 8, most element concentrations were slightly higher in Gucheng than in Beijing. The data statistics showed that the elements with higher concentrations in PM_2.5_ at both sites were Zn, Fe, Pb, and Mn. This occurred because these polluting elements are mainly influenced by the surrounding production and living processes. The Zn contents in Beijing and Hebei were the highest. According to the study, Zn pollution was mainly attributed to the wear of motor vehicle tires and brake pads. The average concentrations of these metal elements in Beijing were 124.96, 66.97, 13.06 and 12.55 ng/m^3^, respectively. In Gucheng, the concentrations were higher than those in Beijing, at 127.25, 81.96, 33.65 and 18.72 ng/m^3^, respectively. The Cu, As, Ni, Cr, Cd and Co concentrations were relatively low. The concentrations of these elements in Beijing were 5.52, 4.72, 1.02, 1.14, 0.48, and 0.08 ng/m^3^, respectively, and in Gucheng, the concentrations of these elements were 8.30, 10.84, 2.19, 2.79, 1.44, and 0.14 ng/m^3^, respectively. This may be due to the higher industrial activity around Gucheng.

### 3.2. PM_2.5_ Source Analysis

To further study the pollution characteristics of PM_2.5_ in Beijing and the surrounding regions in the summer, source analysis of the PM_2.5_ fractions was conducted. In this study, PCA of the carbonaceous fraction and water-soluble ion concentrations (OC, EC, Na^+^, NH_4_^+^, K^+^, Mg^2+^, Ca^2+^, Cl^−^, NO_3_^−^, and SO_4_^2−^) of PM_2.5_ at the monitoring stations in Beijing and Gucheng was performed using SPSS 26.0.

The data measured at the Beijing and Gucheng monitoring sites were imported into SPSS for PCA to obtain the results presented in Table 4 and Table 5. Table 4 shows that ten principal components were extracted, but only principal component 1 and principal component 2 have eigenvalues greater than 1, namely, 5.526 and 1.985, respectively. The cumulative contribution is 72.4%. This indicates that these two principal components can explain most of the data and reflect the main pollution sources in Beijing. Similarly, Table 5 shows that 10 principal components were also extracted for Gucheng with only principal component 1 and principal component 2 having eigenvalues greater than 1, namely, 6.542 and 1.817, respectively. The cumulative contribution is 76.0%. This indicates that these two principal components can explain most of the data and reflect the main pollution sources around Gucheng.

Table 6 shows that the eigenvalues of principal component 1 are high in Beijing and Gucheng. The correlation coefficients of NH_4_^+^, SO_4_^2−^, NO_3_^−^, K^+^, and EC in Beijing’s atmospheric PM_2.5_ were high in principal component 1. The correlation coefficients were 0.959, 0.917, 0.857, 0.848, and 0.802: all above 0.8. OC, Mg^2+^, and Ca^2+^ had high correlation coefficients of 0.765, 0.677, and 0.673 in principal component 2, respectively. The highest correlation coefficients of OC, K^+^, NH_4_^+^, Mg^2+^, and Na^+^ were 0.967, 0.900, 0.894, 0.867, and 0.795, respectively, in principal component 1 of atmospheric PM_2.5_ in Gucheng. The correlation coefficients of Cl^−^, Ca^2+^ and EC in principal component 2 were 0.708, 0.671, and 0.389, respectively. These results show that there are good correlations between all the above ions and the principal components.

The ions with high correlation coefficients with principal component 1 in Beijing in the summer are NH_4_^+^, SO_4_^2−^, NO_3_^−^, K^+^, and EC, indicating that the main component of air pollution in the Beijing urban area is SNA. Therefore, principal component 1 represents the secondary component, biomass combustion, and vehicle exhaust emissions. The ions with higher correlation coefficients with principal component 2 are OC, Mg^2+,^ and Ca^2+^, so principal component 2 represents dust and coal combustion. The ions in the atmosphere of Gucheng with high correlations with principal component 1 are OC, K^+^, NH_4_^+^, Mg^2+^, and Na^+^, so principal component 1 may represent industrial coal combustion, dust, and biomass fuel combustion. The ions with high correlation coefficients with principal component 2 are Cl^−^, Ca^2+^, and EC. Therefore, principal component 2 represents coal combustion and dust.

In summary, the results of the PCA show that the air pollution in Beijing in summer mainly comes from vehicle exhaust emissions and secondary components. Wang et al. [37] used PCA to analyze carbonaceous aerosols in Beijing in the summer and found that the contribution of motor vehicle emissions reached more than 60%. The air pollution in Gucheng mainly comes from industrial coal combustion, dust, and biomass combustion. The two sites have a relatively clear multisource and multipathway composite pollution phenomenon. In general, our results are consistent with previous studies, and secondary sources, vehicle emissions, and regional transport were the main contributors to PM_2.5_ pollution in Beijing [38].

### 3.3. Oxidative Potential Analysis

The AA method was used to determine the OP of the PM_2.5_ samples collected from Beijing and Gucheng. The average OP values in Beijing and Gucheng were 91.6 ± 42.1 and 82.2 ± 47.1 pmol/(min·m^3^), respectively. Figure 9a,b compares the daily PM_2.5_ concentrations and OP values in Beijing and Gucheng, respectively. The trends of Beijing and Gucheng are roughly the same, but the average OP value is slightly higher in Beijing than in Gucheng. Considering the ambient air quality index of China and the PM_2.5_ concentration in these measurements, days with average PM_2.5_ values below 35 μg/m^3^ are defined as clean days, days with values above 35 μg/m^3^ and below 75 μg/m^3^ are defined as lightly polluted days, and days with values above 75 μg/m^3^ are defined as polluted days. The counting results showed that the average OP values on lightly polluted days in Beijing and Gucheng were 120.0 and 106.4 pmol/(min·m^3^), respectively. These values were 1.59 times and 1.5 times greater than the values on clean days (75.3 and 70.7 pmol/(min·m^3^)). As shown in Figure 10, the OP values in Beijing and Gucheng show a trend of increasing with increasing PM_2.5_ concentration. To further investigate the contributions of the different chemical components of PM_2.5_ to the OP in Beijing and Gucheng, Pearson correlation coefficients were used to analyze the correlations between the PM_2.5_ components and OP values using SPSS 26.0.

The Pearson correlation coefficients between the chemical components and the OP of PM_2.5_ at the two locations are shown in Figure 11. In Beijing, the water-soluble ions Cl^−^ and NO_3_^−^ in PM_2.5_ were moderately correlated (0.4 ≤ r <0.6, *p* < 0.01) with the OP. This may be due to the significant contribution of motor vehicle emissions as a source of PM_2.5_ in Beijing. In Gucheng, K^+^ had a strong correlation (r = 0.61) with the OP. Na^+^, SO_4_^2−^, and Mg^2+^ had a moderate correlation with the OP. The correlation of K^+^ may originate from the contribution of biomass burning to PM_2.5_ in Gucheng [39]. SO_4_^2−^ is influenced by its gaseous precursor SO_2_, which may be associated with the more frequent industrial activities around Gucheng. Mg^2+^ is mainly from dust generated by construction activities. The EC in Gucheng PM_2.5_ is also moderately correlated with the OP, which may originate from the fossil fuel consumption caused by industrial activities and vehicle exhaust in the area.

The contribution of water-soluble elements to OP is important, and it can induce the production of ROS [11]. In Beijing, there is a strong correlation between Cu (r = 0.63) and the generation of OP. Co and Ni have a moderate correlation with the OP. These elements may come from vehicle exhaust and brake wear [40]. In Gucheng, Cu, Pb, Mn, and Sb had strong correlations (r = 0.85, r = 0.67, r = 0.65, r = 0.62, respectively) with the OP. Ba, Fe, Cd, Sn, Zn, and Cr show moderate correlations with the OP (r > 0.5). Previous studies have shown that Cu, Pb, Fe, and Zn come from vehicle exhaust and industrial activities [41]. Mn, Sb, Ba, Cd, Sn, and Cr may originate from industrial activities and coal combustion [40]. Hence, industrial activities, vehicle exhaust, and coal combustion may be the main contributing sources to the OP in Gucheng.

The relationships between the chemical components and the OP of PM_2.5_ in Beijing and Gucheng are very different. This mainly occurs because the PM_2.5_ sources are different between the two regions, and the contributions of each component to the PM_2.5_ concentrations are different, resulting in different contributions to the production of OP. The OP of PM_2.5_ in Beijing is mainly affected by vehicle exhaust, while that in Gucheng is also affected by dust and combustion sources, such as industrial coal burning and biomass burning.

### 3.4. Health Risk Assessment

Noncarcinogenic risk calculations were performed for Cu, Zn, Pb, and Mn in the samples collected in Beijing and Gucheng. The risk assessment value of each element was calculated according to Equations (2)–(5). Carcinogenic risk assessment calculations were performed for Cr, Ni, Cd, and As. The results are shown in Figure 12 and Figure 13. Figure 12 shows that for all populations, the noncarcinogenic risk values of Pb and Mn were high, and in Beijing, the HQ values of Pb and Mn for children and adolescents, adult females, and adult males were (5.73 ± 3.37) × 10^−3^, (7.13 ± 4.18) × 10^−3^, (7.49 ± 4.39) × 10^−3^, (7.91 ± 3.57) × 10^−3^, (9.82 ± 4.43) × 10^−3^, and (1.03 ± 0.47) × 10^−3^, respectively. In Gucheng, the HQ values of Pb were higher than those in Beijing, at (1.48 ± 1.11) × 10^−2^, (1.84 ± 1.38) × 10^−2^, and (1.93 ± 1.45) × 10^−2^, respectively. The HQ values of Cu and Zn at both locations were less than 5 × 10^−3^. However, the HQ values of the noncarcinogenic risk factors are all less than 1. Therefore, the noncarcinogenic risks of Cu, Zn, Pb, and Mn are negligible.

The dashed line in Figure 13 denotes R = 10^−6^, which shows that Cr and As exceed this value in Beijing and that Cr, Cd and As exceed this value in Gucheng. Please refer to the determination of the R value in Section 2.4. The carcinogenic risk factors for Ni and Cd in Beijing are less than 10^−6^ for children, adolescents and adults, so the risk is negligible. The cancer risk factor values for Cr and As are between 10^−6^ and 10^−4^, indicating a potential risk for children, adolescents, and adults. The cancer risk factors for adults are higher than those for children and adolescents. The risk assessment calculations in Gucheng are higher than those in Beijing. The risk factor for Ni is less than 10^−6^ for all populations, and therefore, the risk of Ni-induced cancer is negligible. The risk factor for Cd is less than 10^−6^ for children and adolescents. Adults are in the range of 10^−6^ and 10^−4^, representing a potential cancer risk. The carcinogenic risk factor values for Cr and As are between 10^−6^ and 10^−4^, indicating a potential risk for all populations, similar to the situation for Beijing. The risk factors for adults are higher than those for children and adolescents. In summary, Cr and As have potential carcinogenic risks for all populations at both sites, while Cd has a potential carcinogenic risk for adults in Gucheng.

The experimental results are similar to the PM_2.5_ health risk assessment results of Li et al. for 29 provincial capitals in China [42], with Cr and As posing higher carcinogenic risks.

## 4. Conclusions

In this study, we found that Beijing and Gucheng exhibited high concentrations of OC, EC, SO_4_^2−^, NO_3_^−^, and NH_4_^+^ in PM_2.5_. The highest NO_3_^−^ content was 23.23% in Beijing, and the highest SO_4_^2−^ content was 25.09% in Gucheng. Then, we employed PCA and determined that the main sources of PM_2.5_ in Beijing were motor vehicle emissions and secondary components, while industrial emissions dominated in Gucheng. Moreover, both areas were influenced by airflows from the eastern, southern and southwestern directions. The AA method was used to analyze the OP values at the two sites. The contributions of Cu, Cl^−^, and NO_3_^−^ to the OP values were larger in Beijing, and the contributions of Cu, K^+^, Pb, and SO_4_^2−^ were notable. This is related to the main sources of PM_2.5_ at both locations. The health risk assessment results indicated potential carcinogenic risks of Cr and As for all populations at both sites and a potential carcinogenic risk of Cd for adults in Gucheng.

In the future, when studying or developing pollution prevention and control measures in the future, more attention should be given to the health effects of motor vehicle emissions in Beijing and industrial emission control in Hebei, and the effects of regional transmission should be fully considered. In this paper, the source of secondary components is still unknown, to better explore the source of all components, chemical model combined with different seasons experiment results still needs to be further conducted.

## Figures and Tables

**Figure 1 ijerph-20-05109-f001:**
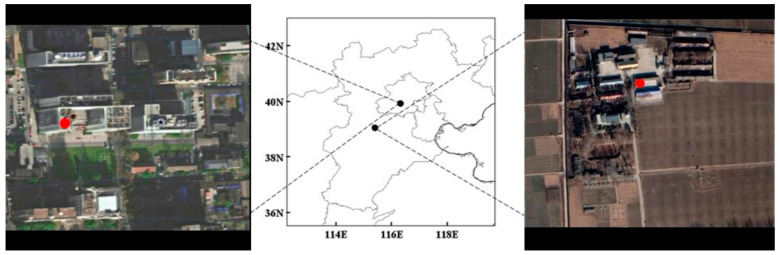
Geographical location of the sampling sites: Beijing (**left**) and Gucheng (**right**).

**Figure 2 ijerph-20-05109-f002:**
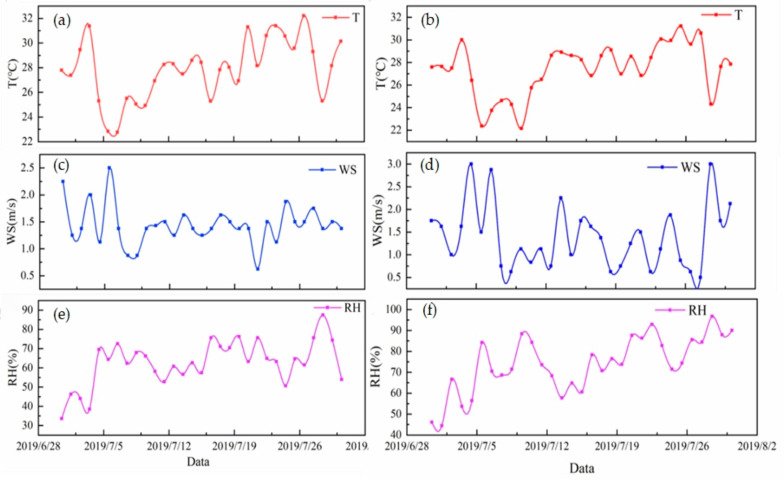
Meteorological conditions in Beijing (**left**) and Gucheng (**right**) during the sampling period. (**a**,**c**,**e**) show time series plots of the temperature (T), wind speed (WS) and relative humidity (RH), respectively, in Beijing. (**b**,**d**,**f**) show time series plots of the T, WS, and RH, respectively, in Gucheng.

**Figure 3 ijerph-20-05109-f003:**
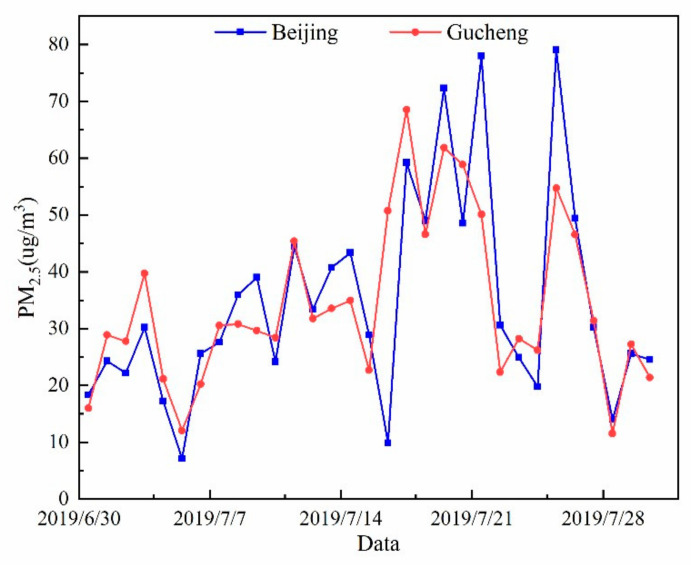
Daily average PM_2.5_ concentrations in Beijing and Gucheng during the sampling period.

**Figure 4 ijerph-20-05109-f004:**
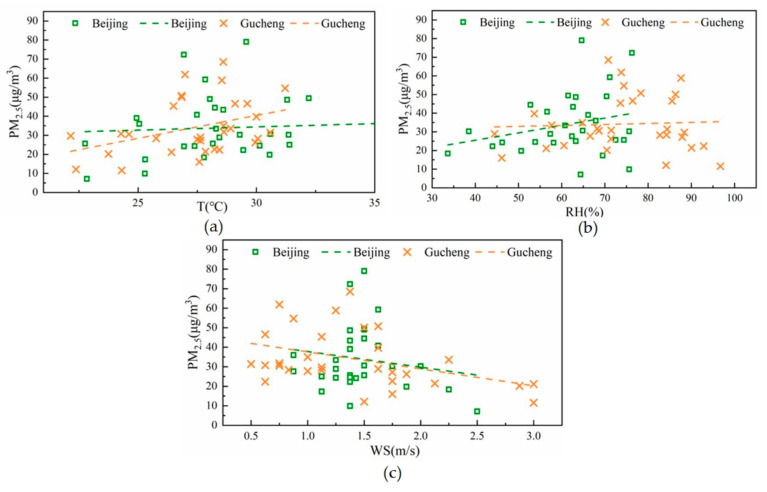
Correlations between the meteorological parameters and PM_2.5_ concentrations at the two sampling sites ((**a**), (**b**), and (**c**) correlate with T, RH, WS and PM2.5, respectively, the green symbols denote Beijing, and the orange symbols denote Gucheng).

**Figure 5 ijerph-20-05109-f005:**
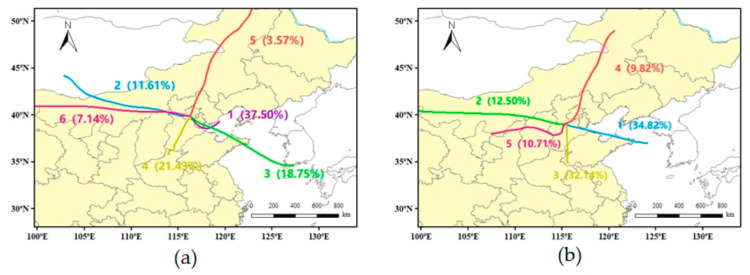
Clustering analysis of the 72 h backward trajectories of the sampling sites in Beijing (**a**) and the surrounding area (**b**).

**Figure 6 ijerph-20-05109-f006:**
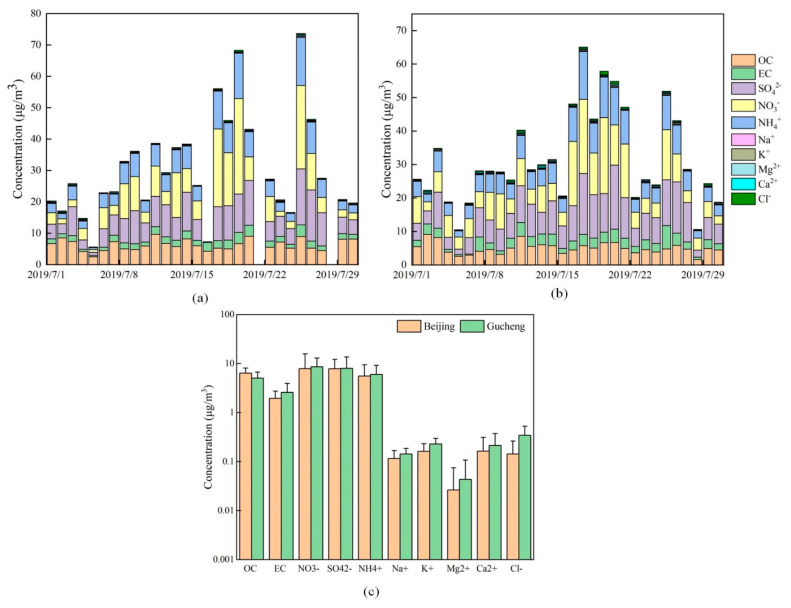
Carbonaceous fraction and water-soluble ion concentrations in Beijing and Gucheng ((**a**,**b**) are the accumulation maps of water-soluble ion concentrations in Beijing and Gucheng, respectively, and (**c**) is the concentrations of water-soluble ions in the two places).

**Figure 7 ijerph-20-05109-f007:**
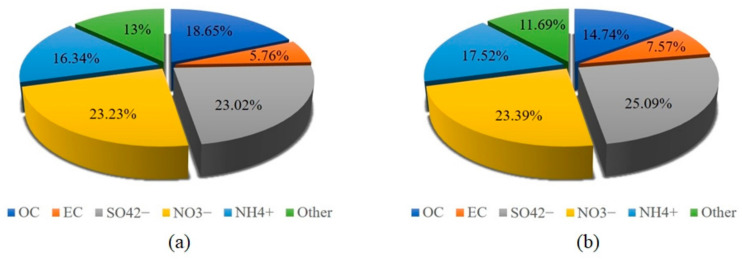
Percentage of each component in Beijing (**a**) and Gucheng (**b**).

**Figure 8 ijerph-20-05109-f008:**
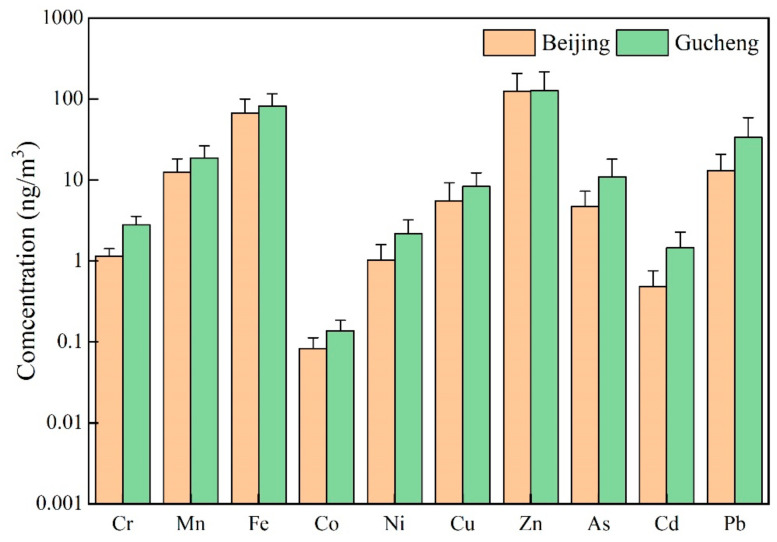
The average concentrations of the water-soluble elements in PM_2.5_ in Beijing and Gucheng.

**Figure 9 ijerph-20-05109-f009:**
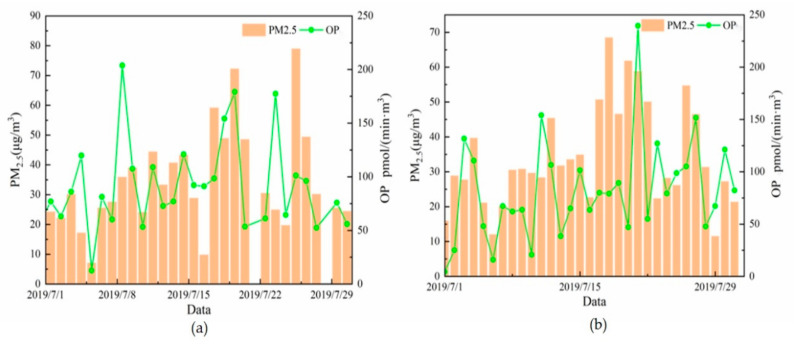
Time series of the OP values and PM_2.5_ concentration in Beijing (**a**) and Gucheng (**b**).

**Figure 10 ijerph-20-05109-f010:**
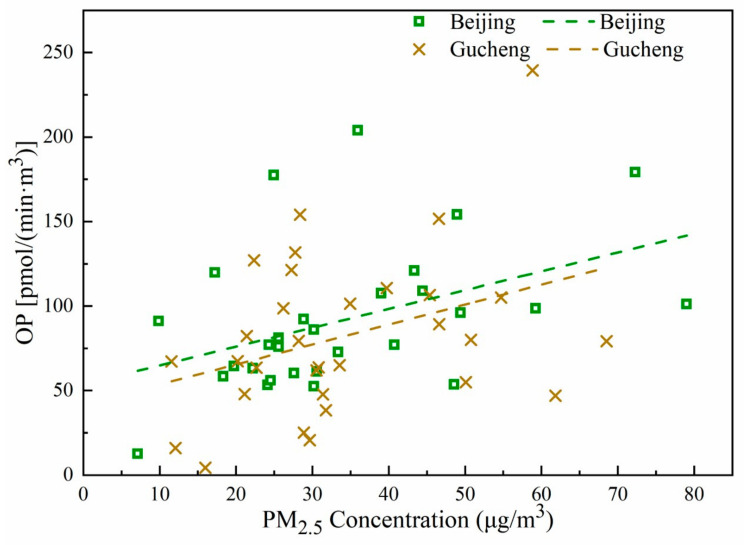
Daily OP values versus PM_2.5_ concentrations in Beijing and Gucheng, along with trendlines.

**Figure 11 ijerph-20-05109-f011:**
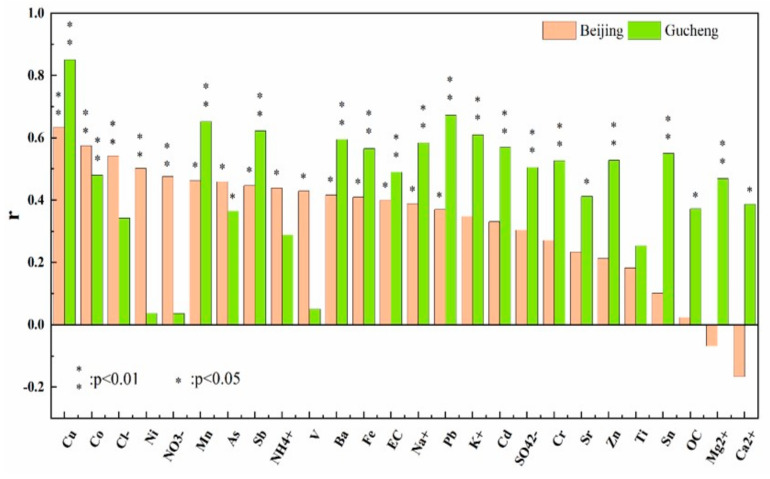
Pearson correlation coefficients between the PM_2.5_ chemical components and oxidative potential in Beijing and Gucheng. (** indicates that the *p*-value of the Pearson correlation is less than 0.01; * indicates that the *p* value is less than 0.05).

**Figure 12 ijerph-20-05109-f012:**
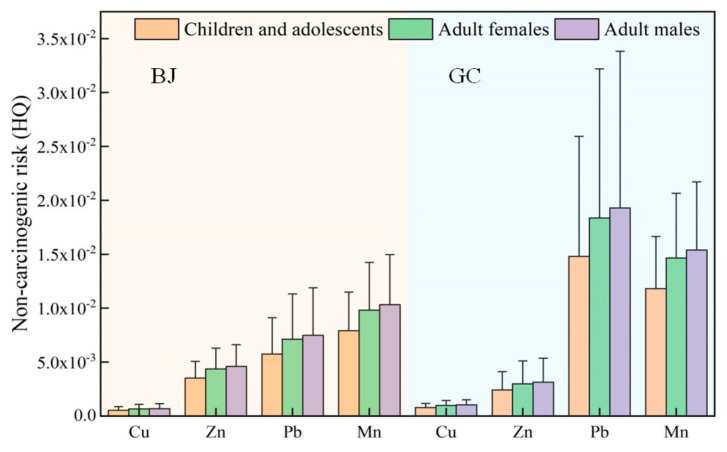
Noncarcinogenic risk factors for the heavy metals in PM_2.5_ in Beijing and Gucheng.

**Figure 13 ijerph-20-05109-f013:**
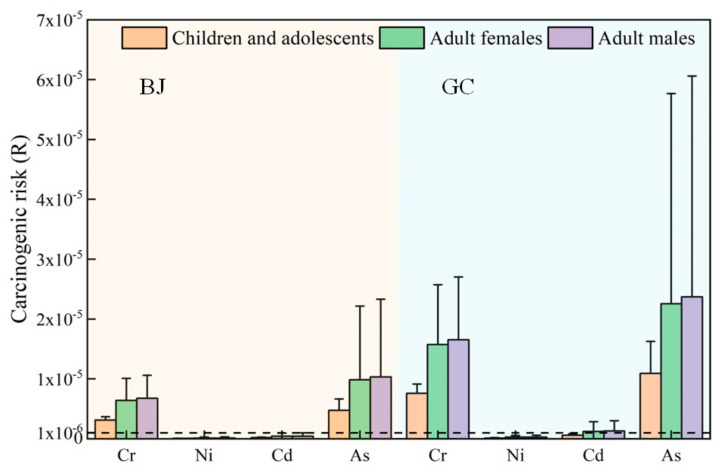
Carcinogenic risk coefficients of the elements in PM_2.5_ in Beijing and Gucheng.

**Table 1 ijerph-20-05109-t001:** Exposure assessment values.

Population	Age	IR (m^3^/d)	EF (d/a)	ED (d)	BW (kg)	AT for Noncarcinogenic Materials (d)	AT for Carcinogenic Materials (d)
Children and adolescents	6−17	8.7	365	18	46	18 × 365	70 × 365
Adult females	≥18	13.5	365	30	57.5	30 × 365	70 × 365
Adult males	≥18	16.6	365	30	67.3	30 × 365	70 × 365

IR: inhalation rate; EF: exposure frequency; ED: exposure duration; BW: body weight; AT: average contact time.

**Table 2 ijerph-20-05109-t002:** Dose-response parameters of the heavy metals.

Element	Nature	RfD (mg/(kg·d))	SF (mg/(kg·d))
Cu	Noncarcinogenic	2.0 × 10^−3^	
Zn	Noncarcinogenic	1.0 × 10^−2^	
Pb	Noncarcinogenic	4.3 × 10^−4^	
Mn	Noncarcinogenic	3.0 × 10^−4^	
Cr	Carcinogenic		56
Ni	Carcinogenic		1.19
Cd	Carcinogenic		8.4
As	Carcinogenic		20.7

RfD: reference dose; SF: carcinogenic intensity coefficient of a given carcinogenic pollutant.

**Table 3 ijerph-20-05109-t003:** Mean PM_2.5_ concentrations for each cluster.

Cluster	PM_2.5_ (μg/m^3^)	
	Beijing	Gucheng
1	35.1	40.7
2	31.0	25.0
3	43.3	34.3
4	48.6	17.4
5	7.1	43.5
6	27.0	

**Table 4 ijerph-20-05109-t004:** Principal component analysis of the variance decomposition results of the PM_2.5_ pollution components in Beijing.

Principal Component	Initial Eigenvalue			Sum of Squares of the Extracted Loads		
	Eigenvalue	Contribution/%	Cumulative Contribution/%	Eigenvalue	Contribution/%	Cumulative Contribution/%
1	5.5256	52.556	53.556	5.5256	52.556	52.556
2	1.985	19.845	72.401	1.985	19.845	72.401
3	0.872	8.720	81.120			
4	0.693	6.931	88.051			
5	0.621	6.213	94.264			
6	0.307	3.074	97.338			
7	0.191	1.911	99.248			
8	0.050	0.503	99.752			
9	0.025	0.246	99.998			
10	0.000	0.002	100.000			

**Table 5 ijerph-20-05109-t005:** Principal component extraction by variance decomposition of the PM_2.5_ pollution components in Gucheng.

Principal Component	Initial Eigenvalue			Sum of Squares of the Extracted Loads		
	Eigenvalue	Contribution/%	Cumulative Contribution/%	Eigenvalue	Contribution/%	Cumulative Contribution/%
1	6.542	59.477	59.477	6.542	59.477	59.477
2	1.817	16.516	75.993	1.817	16.516	75.993
3	0.852	7.742	83.735			
4	0.827	7.514	91.249			
5	0.331	3.009	94.258			
6	0.298	2.713	96.972			
7	0.167	1.519	98.491			
8	0.091	0.832	99.322			
9	0.074	0.671	99.993			
10	0.001	0.007	100.000			

**Table 6 ijerph-20-05109-t006:** Principal component analysis of the Beijing and Gucheng PM_2.5_ carbonaceous fractions and water-soluble ions.

Component	Beijing		Gucheng	
	Principal Component 1	Principal Component 2	Principal Component 1	Principal Component 2
OC	0.360	0.765	0.967	−0.226
EC	0.802	0.035	0.599	0.389
Na^+^	0.777	0.225	0.795	0.352
NH_4_^+^	0.959	−0.178	0.894	0.006
K^+^	0.848	0.451	0.900	−0.390
Mg^2+^	−0.238	0.677	0.867	0.163
Ca^2+^	−0.072	0.673	0.463	0.671
Cl^−^	0.777	−0.332	0.361	0.708
NO_3_^−^	0.857	−0.300	0.733	−0.190
SO_4_^2−^	0.917	0.038	0.732	−0.556

## Data Availability

The data sets used and/or analyzed during the current study are available from the corresponding author on reasonable request.

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
