# Peer review of "Oxidative Potential Characterization of Different PM_2.5_ Sources and Components in Beijing and the Surrounding Region"

_ijerph, 2023, doi:10.3390/ijerph20065109_

Round 1

Reviewer 1 Report

This manuscript deals with an important topic of particulate matter pollution. However, the presentation of the manuscript needs extensive editing as follows:

1.      Highlights: remove all abbreviations and keep only the full term.

2.      Introduction is very long and needs to be more focused. Also, several long paragraphs have been written without reference (E.g. Lines 82-93).

3.      Material and methods:

-          Line 119-126: on what basis have the authors chosen the sampling protocol?

-            Line 143: where is the reference of the modified-A temperature protocol?

-          More details of the metals analysis using an inductively coupled plasma mass spectrometer should be added.

-          The quality control of the analytical technique of metal determination should be described in detail.

-          Table 2: clarify the full term of metals in the footnote of the table.

4.      Results:

-          The data presented in figures 7, 8, 12, and 13 were not statically analyzed. The data should be presented as means±SD. Also, the data should be statistically analyzed to reveal significant differences.

5.      The conclusion section in its present form is just a summary of the results, not a conclusion. The conclusion section needs to be rewritten in a more concise and informative way, reflecting the main findings of the study, practical applications, and future perspectives.

6.      It is not preferable to begin sentences with abbreviations like PM2.5 in line 13.

7.      Throughout the manuscript, revise the formatting of the references E.g. air pollution.[1] should be air pollution [1].

8.      There is a problem with using abbreviations throughout the manuscript. The full term should be mentioned first with the abbreviation between paresis then the abbreviations should be exclusively used throughout the manuscript. E.g., in the abstract, there is no need to give an abbreviation for principal component analysis (PCA) in line 19 as it has not been repeated again.  Also, in lines 32 -33, the particulate matter should be replaced by PM as it has been abbreviated, and the full term has been repeated many times again throughout the manuscript, lines 53, 54, 56, 57, 65, 72, 79, and 175. Such errors have been repeated for many abbreviations throughout the manuscript.

Reviewer 2 Report

This paper is within the scope of the Journal and a timely subject. There are many studies on PMs and their oxidative potential, which does not mean that research in this area should not be conducted. On the contrary, the more such works, the better for learning and expanding knowledge about the impact of PM components on humans. Authors clearly defined the purpose of their work and its scope. The methodology is correct. However, it requires a more detailed description. Results are properly described and discussed. Conclusions are supported by the results and goals. Paper is referenced adequately. English grammar and spelling are fine.

Suggestions:

Correct the citation in the text. There is .[1] and it should be [1].

Unify the notation of PM2.5 in the text of the work -  see Figure 2.

Explain why PM2.5 samples were collected only in the summer.

Complete the information on the limits of detection of the analytical method for ions and metals, description of the method - sample preparation, conditions of chromatographic separation for ions.

Round 2

Reviewer 1 Report

The authors have made significant revisions to the manuscript. However, the statistical analysis of the data still represents a major concern as follows:  the columns in figures 11, 12, and 13 should present the data as means ± SD, not only means. Also, the significance symbols should be added.
